

# Assessment of metrics in next-generation sequencing experiments for use in core-genome multilocus sequence type

Yen-Yi Liu[1,*], Bo-Han Chen[2,*], Chih-Chieh Chen[3] and Chien-Shun Chiou[2]

[1] Department of Public Health, China Medical University, Taichung, Taiwan
[2] Center for Research, Diagnostics and Vaccine Development, Centers for Disease Control, Ministry of Health and Welfare, Taichung, Taiwan
[3] Institute of Medical Science and Technology, National Sun Yat-sen University, Kaohsiung, Taiwan
[*] These authors contributed equally to this work.

## ABSTRACT

With the reduction in the cost of next-generation sequencing, whole-genome sequencing (WGS)–based methods such as core-genome multilocus sequence type (cgMLST) have been widely used. However, gene-based methods are required to assemble raw reads to contigs, thus possibly introducing errors into assemblies. Because the robustness of cgMLST depends on the quality of assemblies, the results of WGS should be assessed (from sequencing to assembly). In this study, we investigated the robustness of different read lengths, read depths, and assemblers in recovering genes from reference genomes. Different combinations of read lengths and read depths were simulated from the complete genomes of three common food-borne pathogens: *Escherichia coli*, *Listeria monocytogenes*, and *Salmonella enterica*. We found that the quality of assemblies was mainly affected by read depth, irrespective of the assembler used. In addition, we suggest several cutoff values for future cgMLST experiments. Furthermore, we recommend the combinations of read lengths, read depths, and assemblers that can result in a higher cost/performance ratio for cgMLST.

## INTRODUCTION

With the reduction in the cost of next-generation sequencing (NGS), whole-genome sequencing (WGS)–based methods are being widely used in genomic epidemiology to characterize bacterial pathogens and perform strain typing (*Deng, Bakker & Hendriksen, 2016*; *Fratamico et al., 2016*; *Lindsey et al., 2016*). Multilocus sequence type (MLST) genotyping (*Maiden et al., 1998*) has been used for many years for cross-laboratory comparison and outbreak investigation among closely-related strains. Core-genome MLST (cgMLST), an advanced version of MLST genotyping, is a genome-wide gene-by-gene comparison approach (*Maiden et al., 2013*) that has been successfully used for detecting disease clusters and investigating outbreaks (*Barkley, Gosciminski & Miller, 2016*; *De Been et al., 2015*; *Jackson et al., 2016*). Several websites and databases, such as PubMLST.org (*Jolley, Bray & Maiden, 2018*) and Pathogenwatch (https://pathogen.watch/), that are

Corresponding author
Chien-Shun Chiou,
nipmcsc@cdc.gov.tw

funded by large companies and governments have been using cgMLST. Because of the increasing significance of cgMLST in the field of epidemiology, evaluating its robustness is crucial. *Segerman (2020)* has reviewed sequencing technologies and assembly methods for the bacterial surveillance and the RefSeq Genome Database. He found that Illumina sequencers were the mostly used sequencing platforms and SPAdes, SKESA and CLC were the most popular assemblers. Based on *Segerman*'s (*2020*) findings, we designed a metric "number of core genes unrecalled" to find out the minimum sequencing depth/coverage for SPAdes, SKESA and CLC at read lengths with 150 bp and 250 bp, which were common in Illumina platforms, to recover the most completely "core gene alleles" (*i.e.,* not only gene locus but also nucleotide sequence of such gene locus needed to be the same). The idea of metric "core gene unrecalled" was from the benchmarking metrics of genome assemblies (*i.e.,* Contiguity, Correctness and Completeness) suggested by *Molina-Mora et al. (2020)* with the scale from genome level down to gene level. Also, because the genes order within a genome does not influence the generated cgMLST profile, we only consider the correctness and completeness of core genes. Therefore, our designed metric "number of core genes unrecalled" could fully reflect the quality of cgMLST profiles. Since the sequencing read length, read depth, and assembler might substantially affect cgMLST results, we investigated the effect of these factors on cgMLST results. In this study, we simulated different read depths of different lengths from four common food-borne pathogens, namely, *Escherichia coli*, *Listeria monocytogenes*, and *Salmonella enterica*, and performed assembling by using different assemblers to determine the minimum read depths required under different situations (*i.e.,* different combinations of read lengths, read depths, and assemblers). The minimum read depths determined in this study might help researchers in estimating the depths before conducting cgMLST studies.

## METHODS AND MATERIALS

To evaluate the minimum read depth required for recalling genes, we simulated read sets with different read depths from complete reference genomes downloaded from NCBI. Three food-borne pathogens were tested: *E. coli*, *L. monocytogenes*, and *S. enterica*. Different assemblers and read lengths were included in the evaluation. The experiments were repeated three times to ensure the robustness of the results.

### Bacterial genomes used for evaluation

IAI39 (*Touchon et al., 2009*), EGD-e (*Toledo-Arana et al., 2009*), and LT2 (*McClelland et al., 2001*), which were the NCBI reference genomes with complete assembly level, were selected for representing *E. coli*, *L. monocytogenes*, and *S. enterica*, respectively. The art_illumina simulator of ART simulation toolkit (*Huang et al., 2012*) was used to generate pseudo reads with different read lengths and read depths from the selected four complete genomes. The command of art_illumina used in this research is "art_illumina –p –na –ss MSv3 -i <reference>-l <read length>-f <depth>-m <read length + 50>-s 10 –o <path/file>".

### Metrics used for evaluation

The metric "number of core genes unrecalled (*i.e.,* number of void cgMLST loci or error called cgMLST alleles)" was designed for finding out the minimum sequencing

depth/coverage for SPAdes, SKESA and CLC at common Illumina produced read lengths of 150 bp and 250 bp to recover the most completely "core gene alleles", which means exactly the same with core gene sequences. In addition, because the genes order within a genome does not influence the generated cgMLST (*Maiden et al., 2013*) profile, we only consider the correctness and completeness of core genes. Therefore, the quality of cgMLST profiles could be reflected through evaluating "number of core genes unrecalled". The cgMLST allele calling was achieved by using BENGA server (*Chen et al., 2021*).

## Evaluation of the minimum sequencing depths achieving stable number of core genes unrecalled by using different assemblers for different read lengths

To evaluate read depths required for different read lengths, we simulated 14 sequencing depths or coverages ($10\times, 20\times, 30\times, 40\times, 50\times, 60\times, 70\times, 80\times, 90\times, 100\times, 200\times, 300\times$, $400\times$, and $500\times$) from *S. enterica* LT2, *E. coli* IAI39, and *L. monocytogenes* EGD-e. Each simulated read set was assembled using SPAdes (*Bankevich et al., 2012*), CLC Genomics Workbench v10.1.1 (CLC), and SKESA (*Souvorov, Agarwala & Lipman, 2018*), and the resulting contigs were compared with the original complete genomes. The reads assembly settings for the three assemblers were listed in Table S1. All genes were predicted using the Prodigal program (*Hyatt et al., 2010*). The "gene recalled" was defined as the predicted gene in the assembly showed a 100% match with the predicted gene in the original complete genome. The three assemblers used for the read lengths of 150 and 250 bp were compared to determine the minimum coverage needed to recover the maximum genes for different read lengths, regardless of assemblers. Deviations in the number of unrecalled genes for the same assembler, read depth, and read length can be caused due to the stochastic procedure of read simulation.

## Evaluation of minimum sequencing depths for the three common food-borne pathogens (*S. enterica*, *E. coli*, *L. monocytogenes*) based on real sequenced data

To reflect the real sequenced reads condition, we picked up genomes both having raw reads data in SRA database and assembled genomes with complete level in GenBank for further evaluation. The complete assembled genome from GenBank can be used as the reference for evaluating the raw reads assembling from SRA. We sampled different read depths (*i.e.,* $10\times, 20\times, 30\times, 40\times, 50\times, 60\times, 70\times, 80\times, 90\times$, and $100\times$) using Seqtk (https://github.com/lh3/seqtk/blob/master/README.md) from the real sequenced reads data of *S. enterica* (SRR5866640 for 150 bp and SRR6929558 for 250 bp), *E. coli* (SRR6924239 for 150 bp and SRR3205757 for 250 bp), and *L. monocytogenes* (SRR3089759 for 150 bp and SRR6347431 for 250 bp). To investigate the minimum read depth required for achieving the stable core genes unrecalled of real sequenced reads data, we picked up the relevant (*i.e.,* having the same BioSample Accession Number) assembled genomes with complete level as the reference for the evaluation. The relevant assembled genomes are *S. enterica* (CP023508.1 for 150 bp and CP036165.1 for 250 bp), *E. coli* (CP029239.1 for 150 bp and CP034799.1 for 250 bp), and *L. monocytogenes* (CP013919.1 for 150 bp and

CP025565.1 for 250 bp). The command for performed Seqtk is "seqtk sample -s [seed] [input] [fraction] >[output]".

**Estimation of the sequencing depth for three commonly used assemblers for completing the assembly process in a linear time**
To evaluate the running time of SPAdes, CLC, and SKESA assemblers, we determined the time required for assembling simulated reads with a read depth of $10\times$, $20\times$, $30\times$, $40\times$, $50\times$, $60\times$, $70\times$, $80\times$, $90\times$, and $100\times$. The read length of 250 bp was chosen for testing. The server equipped with Intel Xeon CPU E7-4830 v4 2.00 GHz was used for the evaluation. The experiment was performed under the condition of eight threads in a 32-GB RAM computation environment. Wall time was used to evaluate the running time.

## RESULTS

We evaluated 14 sequencing depths or coverages ($10\times$, $20\times$, $30\times$, $40\times$, $50\times$, $60\times$, $70\times$, $80\times$, $90\times$, $100\times$, $200\times$, $300\times$, $400\times$, and $500\times$) for determining the assembly quality. The number of unrecalled genes from the reference genomes of *S. enterica* LT2, *E. coli* IAI39, and *L. monocytogenes* EGD-e represented the assembly quality. Three commonly used assemblers, namely SPAdes, CLC, and SKESA, were applied to run the tests. Two read lengths, 150 bp and 250 bp, representing the widely used sequencing read lengths in Illumina HiSeq and Illumina MiSeq platforms, respectively, were evaluated.

**Minimum sequencing depth achieving stable number of core genes unrecalled for different assemblers by using different read lengths**
As shown in Fig. 1, a sequencing coverage of $60\times$ might be a safe choice irrespective of the assembler and read length. We observed that the SPAdes assembler required $30\times$ read depths (irrespective of whether the read length of 150 or 250 bp was used) to achieve minimum depth of the stable core genes unrecalled compared with CLC and SKESA that required read depths of $40\times \sim 60\times$. Because the read lengths of 150 and 250 bp are mainly used in Illumina platforms, we evaluated the minimum sequencing coverage required for these two read lengths. As shown in Fig. 1, sequencing coverages of at least $60\times$ and $50\times$ were required for the read lengths of 150 and 250 bp, respectively, for assembly to achieve the stable core genes unrecalled irrespective of the assemblers used (*i.e.,* SPAdes, CLC, and SKESA). Regarding assemblers, we observed that SPAdes was not considerably affected by the read length and required a read depth of only $30\times$ to recover reference genes. However, CLC and SKESA required a read depth of at least $40\times-50\times$ and $50\times-60\times$, respectively, to achieve assembly quality similar to that obtained using SPAdes.

**Plausible sequencing depths for the three commonly used assemblers to complete the procedure in linear time**
As shown in Fig. S1, the assembly time required by SKESA and CLC did not change even at a sequence depth of $500\times$. However, for SPAdes, the assembly time increased according to the sequence depth, particularly when it was more than $100\times$. In addition, the assembly time was not affected by the read length for SKESA and CLC; however, for SPAdes, a read length of 150 bp required more time for assembly than a read length of 250 bp.

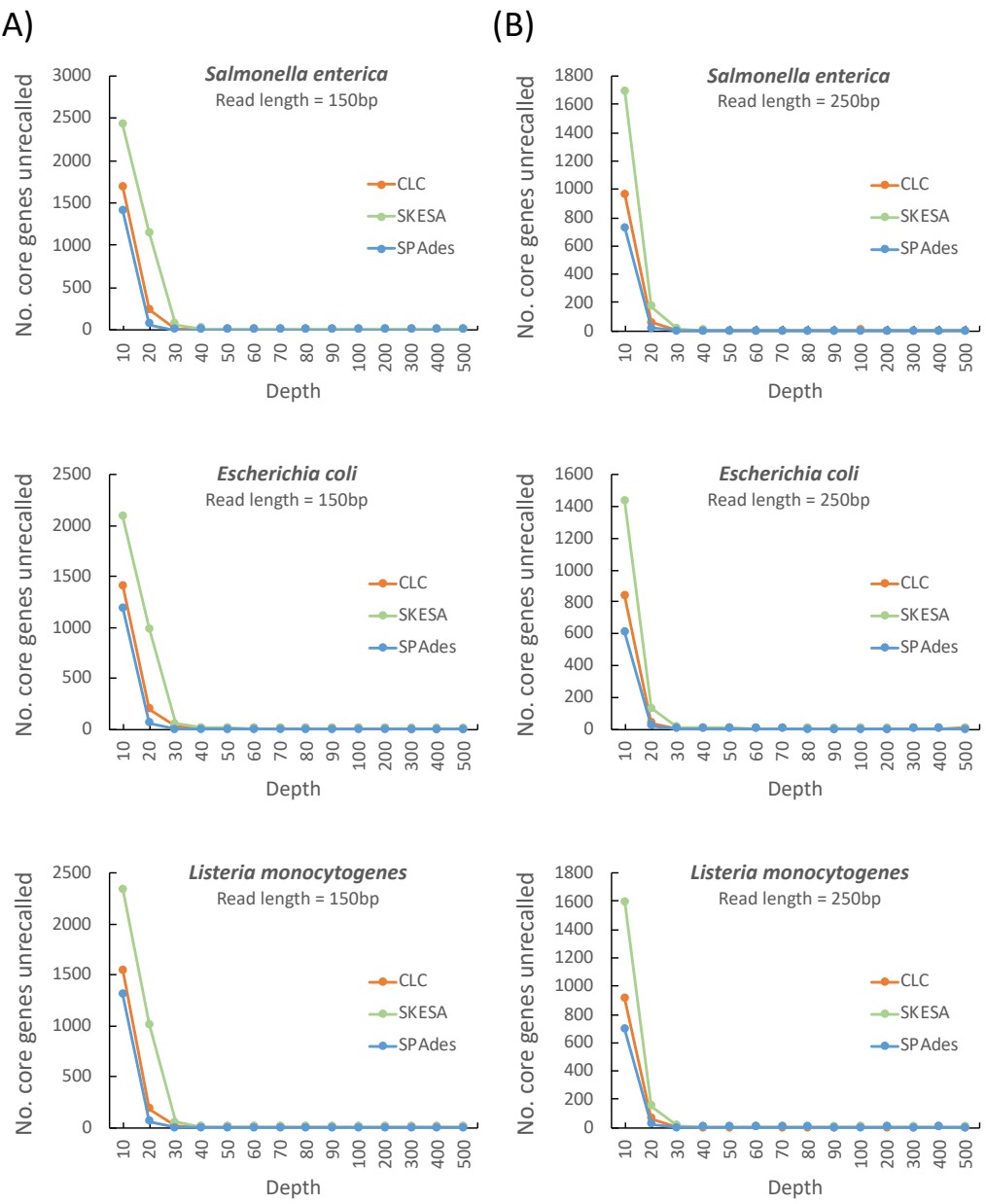

**Figure 1** **Estimation of the minimum read coverage required to achieve the stable core genes unrecalled for assembling at a read length of 150 and 250 bp.** Comparison of different assemblers for the number of unrecalled genes from the reference genome (*S.enterica* LT2, *E. coli* IAI39, and *L. monocytogenes* EGD-e) according to different simulated read coverages (10×, 20×, 30×, 40×, 50×, 60×, 70×, 80×, 90×, 100×, 200×, 300×, 400×, and 500×) at read lengths of 150 (A) and 250 bp (B).

### Suggested minimum sequencing depths for achieving stable number of core genes unrecalled of four common food-borne pathogens (*S. enterica*, *E. coli*, *L. monocytogenes*) based on simulation reads

All the aforementioned evaluation results were obtained from simulated *S. enterica* LT2, *E. coli*, and *L. monocytogenes* reads (shown in Table 1). The minimum required depth tended to be similar among the four tested species, with a minimum depth of 30× for SPAdes and 40×–50× for CLC at a read length of 150 bp and 30× for SPAdes and 40×–50× for CLC at a read length of 250 bp. Compared with SPAdes and CLC, the minimum read coverage for SKESA was 40×–60× at a read length of 150 bp and 40×–50× at a read length of 250 bp. The minimum coverage (depth) of SPAdes, CLC, and SKESA at different read lengths and sequence coverages are highlighted in gray (shown in Table 1).

### Suggested minimum sequencing depths for the three common food-borne pathogens (*S. enterica*, *E. coli*, *L. monocytogenes*) based on real sequenced data

The results of the minimum read depth sampled from real sequenced reads required for achieving the stable number of core genes unrecalled were shown in Table 2. The results were similar to those obtained for simulation data with a minimum depth of 30× for SPAdes and 30×–40× for CLC assemblers at a read length of 150 bp and 20×–40× for SPAdes and 20×–50× for CLC at a read length of 250 bp. Compared with SPAdes and CLC, the minimum read coverage for SKESA was 50×–70× at a read length of 150 bp and 50×–70× for a read length of 250 bp. The reads QC were performed by using FastQC (https://www.bioinformatics.babraham.ac.uk/projects/fastqc/), and the ''Per sequence quality scores'' and ''Sequence Length Distribution'' from QC reports were shown in Fig. S2.

## DISCUSSION

In our evaluation, we applied an index ''genes called'' to represent the assembly quality. Because read sets were simulated from a complete genome, the number of ''genes called'' can directly represent the quality of assemblies. Because the number of genes called can indicate the completeness of a ''pan genome'', which covers ''core genes'', we used genes called as our evaluation index. The three most important factors in NGS were evaluated: read depth, read length, and assembler. We found that an assembler was the most crucial factor that affected the quality of assemblies, especially at a low read depth. For low read depths (20×–30×), SPAdes outperformed CLC and SKESA with an error rate of <2.0%, although the performance of CLC was close to that of SPAdes. Compared with SPAdes and CLC, SKESA usually required 40×–50× to reach an error rate of <2.0%. Although SPAdes demonstrated the highest performance, its running time considerably increased with the read depth, especially when the depth was >100×. No difference in results was observed between long reads (250 bp) and short reads (150 bp) for SPAdes and CLC; however, SKESA required a larger depth to assemble short reads to reach an error rate of <2.0%. In addition, to investigate if some similarity sharing among unrecalled genes at even high sequencing depth, we analyzed the unrecalled core genes of *S. enterica*, *E. coli*

Peerj

**Table 1  The number of core genes unrecalled at each depth comparing for different assemblers based on simulated data.**

| | Read length | Assemblers | Read depth | | | | | | | | | |
|---|---|---|---|---|---|---|---|---|---|---|---|---|
| | | | 10× | 20× | 30× | 40× | 50× | 60× | 70× | 80× | 90× | 100× |
| *S. enterica* (LT2) | 150 bp | SPAdes | 1404–1419 | 64–74 | 3–4[a] | 3–4 | 3 | 3–4 | 3–4 | 3 | 3–4 | 2–3 |
| (Size = 4.9 Mb) | | CLC | 1680–1688 | 241–243 | 10–30 | 3–12 | 2 | 3 | 2–3 | 1–3 | 2–3 | 2–4 |
| | | SKESA | 2382–2443 | 1125–1157 | 62–68 | 13–14 | 8–9 | 5 | 5 | 5 | 5 | 5 |
| | 250 bp | SPAdes | 728–752 | 15–27 | 3–4 | 2 | 1–2 | 1–2 | 1–2 | 2–3 | 1–2 | 2–3 |
| | | CLC | 964–1025 | 55–113 | 5 | 1–2 | 2–3 | 2–3 | 2 | 2 | 2–4 | 2–5 |
| | | SKESA | 1692–1697 | 169–175 | 12–17 | 3 | 2 | 2 | 2 | 2 | 2 | 2 |
| *L. monocytogens* | 150 bp | SPAdes | 1036 | 48 | 0 | 0 | 0 | 0 | 0 | 0 | 0 | 0 |
| (IAI39) | | CLC | 1223–1226 | 139–143 | 3–5 | 0 | 0 | 0 | 0 | 0 | 0 | 0 |
| (Size = 2.9 Mb) | | SKESA | 1839 | 790–791 | 32 | 1 | 0–1 | 1 | 1 | 1 | 1 | 1 |
| | 250 bp | SPAdes | 531–560 | 12–21 | 0–1 | 0 | 0–1 | 0 | 0–1 | 0 | 0 | 0 |
| | | CLC | 686–735 | 36–66 | 0–2 | 0 | 0–2 | 0 | 0 | 0 | 0 | 0 |
| | | SKESA | 1217–1267 | 111–126 | 3–11 | 1–3 | 1 | 1 | 1 | 1 | 1 | 0–1 |
| *E. coli* (EGD-e) | 150 bp | SPAdes | 1188 | 61 | 4 | 4 | 3–4 | 4 | 3–5 | 3 | 2–3 | 3–4 |
| (Size = 4.6 Mb) | | CLC | 1411–1416 | 197–202 | 30–32 | 5–8 | 5–6 | 4–5 | 5 | 2–6 | 5 | 5–6 |
| | | SKESA | 2089 | 989 | 54 | 14 | 9–11 | 8 | 8 | 7–8 | 7 | 7 |
| | 250 bp | SPAdes | 611–629 | 21–22 | 4 | 3 | 2–3 | 3 | 3 | 2–3 | 2 | 2 |
| | | CLC | 811-839 | 40–60 | 4–6 | 3–4 | 3–4 | 2–4 | 3–4 | 4–5 | 2 | 4 |
| | | SKESA | 1403–1432 | 132–135 | 13–14 | 7–8 | 7 | 6–7 | 6–7 | 6 | 6 | 6 |

Notes.

[a] The gray fill represents the minimum read depth needed to achieve the stable number of core genes unrecalled for the combing of different read lengths and assemblers.

Liu et al. (2021), *PeerJ*, DOI 10.7717/peerj.11842

Peer J

**Table 2** **The number of core-gene differences between assembly and the reference genome in each depth comparing for different assemblers based on real sequenced data.**

| | Read length | Assemblers | Read depth | | | | | | | | | |
|---|---|---|---|---|---|---|---|---|---|---|---|---|
| | | | 10× | 20× | 30× | 40× | 50× | 60× | 70× | 80× | 90× | 100× |
| *S. enterica* | **150 bp** | SPAdes | 180–206 | 8–11 | 7–8[a] | 7–8 | 7–8 | 7–8 | 7–8 | 7–7 | 6–8 | 7–8 |
| | (CP023508.1) | CLC | 384–439 | 13–17 | 7–8 | 7–8 | 7–9 | 7–9 | 7–7 | 7–9 | 7–9 | 7–7 |
| | SRR5866640 | SKESA | 2669–2864 | 871–1177 | 89–652 | 24–55 | 13–14 | 12–13 | 13–14 | 13 | 13–14 | 13 |
| | **250 bp** | SPAdes | 185–214 | 11–15 | 8–9 | 8–10 | 8–9 | 8–9 | 8 | 8–9 | 8–9 | 8 |
| | (CP036165.1) | CLC | 338–392 | 12–28 | 10 | 7–11 | 7–8 | 5–9 | 8–9 | 5–8 | 8–9 | 7–8 |
| | SRR6929558 | SKESA | 2373–2570 | 874–885 | 120–145 | 16–22 | 11–12 | 10–11 | 10 | 10 | 10–11 | 10 |
| *L. monocytogens* | **150 bp** | SPAdes | 376–423 | 12–14 | 0–1 | 0 | 0 | 0 | 0 | 0 | 0 | 0 |
| | (CP013919.1) | CLC | 593–658 | 41–60 | 3–5 | 0–2 | 0–1 | 1 | 0–1 | 1 | 1 | 0–1 |
| | SRR3089759 | SKESA | 2000–2059 | 1144–1192 | 660–671 | 55–130 | 6–14 | 2–3 | 1–3 | 1–2 | 1 | 1 |
| | **250 bp** | SPAdes | 176–200 | 7–14 | 1–2 | 0–2 | 0 | 0 | 0–1 | 0 | 0 | 0 |
| | (CP025565.1) | CLC | 325–349 | 40–55 | 7–12 | 3–8 | 0–1 | 0–3 | 1 | 0–1 | 0 | 0 |
| | SRR6347431 | SKESA | 1521–1620 | 597–612 | 101–125 | 22–39 | 6–7 | 3–5 | 1–3 | 0–2 | 0–3 | 1–2 |
| *E. coli* | **150 bp** | SPAdes | 97–122 | 11–12 | 9–12 | 9–10 | 10 | 9–11 | 9–10 | 9–10 | 9 | 9–10 |
| | (CP029239.1) | CLC | 217–234 | 21–26 | 12–17 | 11–14 | 11–13 | 11–13 | 11 | 11–12 | 11 | 11–13 |
| | SRR6924239 | SKESA | 2017–2137 | 905–922 | 468–518 | 34–60 | 17–21 | 12–14 | 13 | 11–12 | 12 | 12 |
| | **250 bp** | SPAdes | 37–56 | 4 | 4 | 4 | 4 | 4 | 4 | 4 | 4 | 4 |
| | (CP034799.1) | CLC | 76–99 | 6 | 5–6 | 5–6 | 5–6 | 4–6 | 5–6 | 5–6 | 5–7 | 6–7 |
| | SRR3205757 | SKESA | 1651–1780 | 449-553 | 22–30 | 7–8 | 6 | 6 | 6 | 6 | 6 | 6 |

Notes.

[a] The gray fill represents the minimum read depth needed to achieve the stable number of core genes unrecalled for the combing of different read lengths and assemblers.

and *L. monocytogenes* at depth 100×, and no commonality among these genes was found. The unrecalled core genes at depth 100× from Table 1 are listed in Table S2 (the union of the triple repeat are listed). In summary, we recommend sequencing at a read depth of 30×–50× and a read length of 250 bp by using SPAdes as the assembler to maintain a balance of cost/pay ratio in cgMLST.

### Funding
This study was supported by the Ministry of Health and Welfare, Taiwan with Grant No. MOHW106-CDC-C-315-114712. The funders had no role in study design, data collection and analysis, decision to publish, or preparation of the manuscript.

### Grant Disclosures
The following grant information was disclosed by the authors:
The Ministry of Health and Welfare, Taiwan: MOHW106-CDC-C-315-114712.

### Competing Interests
The authors declare there are no competing interests.

### Author Contributions
- Yen-Yi Liu conceived and designed the experiments, performed the experiments, analyzed the data, prepared figures and/or tables, authored or reviewed drafts of the paper, and approved the final draft.
- Bo-Han Chen conceived and designed the experiments, performed the experiments, analyzed the data, prepared figures and/or tables, and approved the final draft.
- Chih-Chieh Chen conceived and designed the experiments, performed the experiments, prepared figures and/or tables, authored or reviewed drafts of the paper, and approved the final draft.
- Chien-Shun Chiou conceived and designed the experiments, authored or reviewed drafts of the paper, and approved the final draft.

### Data Availability
The data are available at NCBI SRA: SRR5866640, SRR6929558, SRR3089759, SRR6347431, SRR6924239 and SRR3205757.
The settings for running the assembly are available in Table S1.

### Supplemental Information
Supplemental information for this article can be found online at http://dx.doi.org/10.7717/peerj.11842#supplemental-information.

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
