# Peer review of "Assessment of metrics in next-generation sequencing experiments for use in core-genome multilocus sequence type"

_PeerJ, doi:10.7717/peerj.11842_

## Round 0.1 · original submission · Major Revisions

I concur with both reviewers in that major revisions are in order before this manuscript can be considered further. Please consider that in order to make this contribution more relevant, other metrics should be considered in addition to genes called regarding cgMLST outcomes. In addition, please pay special attention to each of the reviewer’s specific comments and provide your disposition to each item upon resubmission.

·

Basic reporting

In this study, the authors performed benchmark tests of three major genome assemblers, SPAdes, CLC, and SKESA. They used Seqtk to simulate different read lengths and read depths from Campylobacter jejuni NCTC strains 11168, Escherichia coli IAI39, Listeria monocytogenes EGD-e, and Salmonella enterica LT2. They compared the number of the “completeness of core genes" and the time required for assembling simulated reads with several read depth among the three assemblers. As a result, the SPAdes assembler outperformed and required fewer read depths to recover original genes compared with CLC and SKESA. On the other hand, the assembly time increased according to the sequence depth, particularly when it was more than 100×. The assembly time required by SKESA and CLC did not change much even at a sequence depth of 500×.

The manuscript is written in clear English, but lacks in-depth discussions. For example, the authors should mention previous reports such as "Segerman B. Front Cell Infect Microbiol. 2020; 10: 527102.", "Molina-Mora JA, et al. Sci Rep. 2020; 10(1): 1392." and so on in Introduction or Discussion. In addition, this manuscript does not include raw data of simulated read data. Please add the data or more detail of Seqtk setting to allow other researchers to replicate.

More minor point; the image qualities of Fig. 1 and 2 are a little rough.

Experimental design

As stated above, the authors estimated the performance of three assemble programs using simulated read data. The comparative evaluation would be important for many researchers utilizing next generation sequencers.

However, the assemble details such as specified options were not provided. This information would be needed to understand the results concisely. And then, there would be some differences between the simulated data and real sequenced data. The authors would need to take into consideration the difference and explain it. Alternatively, the authors may be able to compare assembled data using public-deposited short sequencing data of bacterial complete genomes.

Validity of the findings

Although they compared completeness of core genes and the time required for assembling, they did not show whether there are significant differences among the three programs. If the authors could perform a statical analysis, the quality of this paper would be enhanced.

They also showed the number of core-gene differences between assembly and the three reference genomes in Table 2. Does the number mean unrecalled genes? I do not understand the reason that they exclude C. jejuni in this table. And then, there were several different genes in S. enterica and E. coli at even 100× read depth. Did these genes have any similarity? Why did L. monocytogenes show limited different gene number? Please add experiments or explanations concerning these questions to the discussion.

Souvorov et al. described "we showed that the quality of SKESA assemblies is better than both SPAdes and MegaHit" (Souvorov A. et al. Genome Biology 2018; 19: 153). Please explain the reason of this discrepancy.

Additional comments

I described my comments and questions above.

Reviewer 2 ·

Basic reporting

In this manuscript, the authors describe how a simple metric, the genes called by a cgMLST tool, varied in simulations of genome sequences with various coverage depth and read lengths. This is an interesting question, but the analyses done are very limited and the results are not always presented clearly in figures or tables, without enough information for the reader to understand quickly.

Use present tense at the end of Abstract (last two sentences)

Lines 88-92 Need to revise sentences structure, these are missing words or are not grammatically correct

Line 111 Not clear here what is considered a threshold “to recover original genes”. Do the authors mean all genes or some proportion are recovered at a certain coverage?

Line 145 What is meant by real data ? The authors need to specific if these are raw reads or quality controlled reads (errors corrected) and indicate some information about average read length, etc.

Experimental design

The most important factor that has not been considered is the quality of reads and the potential of sequencing errors affecting read calling for cgMLST. Genes called is an important metric, but should be considered in conjunction with alleles called.

Validity of the findings

The findings are valid, but more care should be taken in explaining the data used in each simulation (quality of the genomes, level of assembly, raws read used or not, etc).

Figure 2 is superfluous, it contains information that can be summarized in a sentence.

Table 1. It is not immediately clear what metric is used to judge the performance of different assemblers with the two read lenghts

---

## Round 0.2 · Minor Revisions

I agree with this reviewer in that additional modifications are still necessary prior to acceptance.

·

Basic reporting

The manuscript by Liu et al. was improved, but some minor points need to be revised.

#1. Concerning Table 1 and 2, those right-side columns (probably x100?) were lost in the manuscript file. The authors would need to adjust the table or some settings of the file.

#2. (Fig. 1, Table 1, Table 2, and so on) Please add spaces between numbers and units (ex. "150 bp").

#3. The information of unrecalled core genes at depth 100x might be useful for some public health researchers, since the data indicated that there are some difficulties to be recalled in the genes. The authors showed the data in their rebuttal letter. I would like to suggest the authors to include the data in the manuscript or supplementary data.

Experimental design

no comment

Validity of the findings

no comment

Additional comments

I described my comments in "Basic reporting" section.

---

## Round 0.3 · accepted · Accept

I agree with this reviewer in that all modifications have been sufficiently addressed and your manuscript is now ready for acceptance.

·

Basic reporting

no comment

Experimental design

no comment

Validity of the findings

no comment